# MULTI-METHOD SELF-TRAINING: IMPROVING CODE GENERATION WITH TEXT, AND VICE VERSA

## ABSTRACT

Large Language Models have many methods for solving the same problem. This introduces novel strengths (different methods may work well for different problems) and weaknesses (it may be difficult for users to know which method to use). In this paper, we introduce Multi-Method Self-Training (MMST), where one method is trained on the filtered outputs of another, allowing us to augment the strengths and ameliorate the weaknesses of each method. Using a 176B parameter model trained on both language and code, we show that MMST can 1) improve the less performant method (up to 30%) making the model easier to use, 2) improve the more performant method (up to 32.2%) making the model more performant, and 3) improve the performance of related but distinct tasks (up to 10.3%) by improving the ability of the model to generate rationales. We then conduct ablation analyses to explore why MMST works. We show that MMST generates more data than traditional self-training, but the improvement in performance is driven by the use of multiple methods. We also analyze prompt-engineering and anti-correlated performance between methods as means of making MMST more effective. We hope the evidence from our paper motivates machine learning researchers to explore ways in which advances in language models allow for new forms of training.

## 1 INTRODUCTION

As foundational models become more capable, they develop more ways of solving the same problem. This can be seen clearly with multi-modal models. If a model trained on both images and text is given a picture of a Shakespearean play and asked to identify the characters, it could do this in many ways: for example, it could try to directly identify the characters, it could first convert the scene to a textual description then identify the characters from Shakespeare's own descriptions, it could generate many images of each character from Shakespeare's descriptions then identify which has the greatest similarity to the characters in the scene, and so on.

This property, however, extends to other varieties of model as well. Models trained on both text and code can often solve a problem using either means. Indeed, LLMs can solve the same problem using many different prompts, often with widely varied results.

This lends to both the weaknesses and strengths of more complex models. Prompting can be extremely non-obvious, leading to a sub-optimal user experience requiring significant prompt engineering to get the desired results. This has led to many methods attempting to optimize prompts (Li and Liang, 2021; Liu et al., 2021; Lester et al., 2021; Reynolds and McDonell, 2021). On the other hand, different methods of doing the same task might have different strengths, and the best method can be used for the particular task at hand – for example, Chain-of-Thought Prompting (Wei et al., 2022) for reasoning tasks or PAL (Gao et al., 2022) for the kinds of tasks found on BigBench Hard (Srivastava et al., 2022; Suzgun et al., 2022).

In this paper, we propose a method to ameliorate the weaknesses and augment the strengths of increasingly complex models: multi-method self-training (figure 1). We can translate the correct answers from one method into training instances for the other, and then apply existing self-training techniques. Using BLOOM-176B, we demonstrate the effectiveness of this technique by solving math problems using both text generation and code generation, then improving the performance

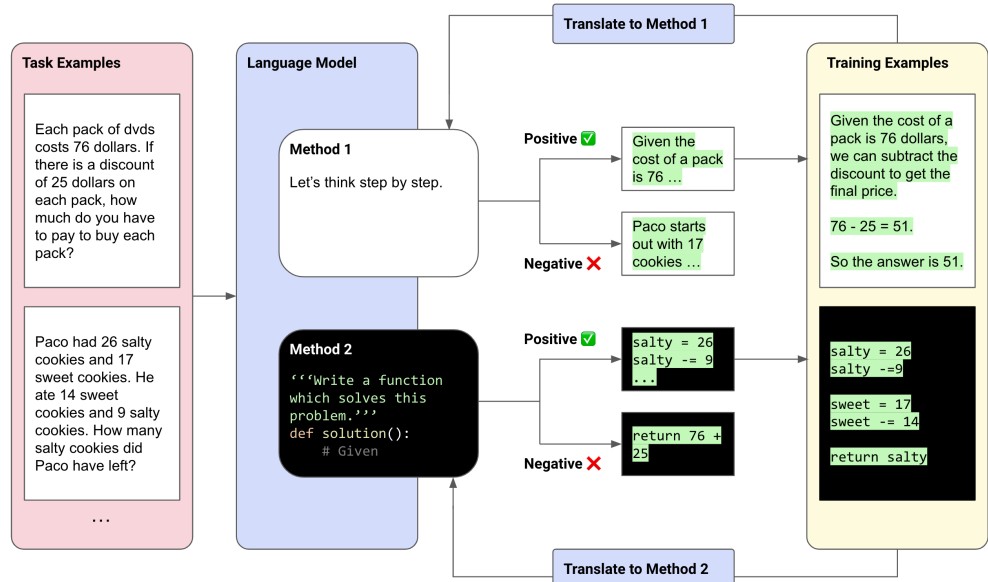

Figure 1: Overview of multi-method self-training. Given a series of task examples (containing only the instructions for a task), the model generates solutions using multiple methods (in this paper, we focus on text and code). The solutions for each method are then assigned a pseudo-label (in this paper, we assign a positive pseudo-label if the model gets the correct answer, although the steps to produce the answer may be wrong). The training examples from both methods are then translated to each method (text examples are turned into code, code examples are turned into text), and then used to finetune each method.

of both methods using multi-method self-training. Multi-method self-training improves the performance of solving math problems via text (23.6% on SVAMP, 27.0% on GSM8K, 30.0% on MAWPS, 4.6% on MATHQA), solving math problems via code (32.2% on SVAMP, 20.1% on GSM8K, 0.0% on MAWPS, 7.6% on MATHQA), and solving out-of-domain reasoning tasks more effectively as well (6.9% on StrategyQA, 10.3% on CommonSenseQA). Lastly, we conduct ablation studies to understand why multi-method self-training works. We hope that the wide applicability of our method, the simplicity of its application, and the strong results encourage further study of the methods by which we can train models.

Our contributions are as follows:

- We demonstrate that multi-method self-training can improve both the less performant method that is being trained (a user experience improvement) and the more performant method that is being trained (an improvement in overall performance).

- We demonstrate that multi-method self-training can improve out-of-domain tasks which are related to the self-training task.

- We provide detailed ablation studies for two hypotheses as to why multi-method self-training works: increased data quantity and anti-correlation between methods. Both contribute to the effectiveness of the method.

## 2 RELATED WORK & BACKGROUND

Self-training is a method for improving models using unlabeled examples. An initial model is used to predict labels for those examples, then a confidence measure is applied to the predicted labels. Predicted labels with high confidence-scores are then used as pseudo-labels to train the model. A survey of self-training techniques can be found in Amini et al. 2022.

The use of a confidence measure is critical – using all the predicted labels as pseudo-labels would result in performance identical to that of the original model (Chapelle et al., 2006). Traditional confidence measures typically rely on the probability provided by the model to each class, with pseudo-labels being chosen as those predicted labels which have probability above a certain threshold (Tür et al., 2005; Zou et al., 2018; Zhang et al., 2021).

More recent methods applying self-training to LLMs have identified a number of alternative confidence measures (Haluptzok et al., 2022; Zelikman et al., 2022; Huang et al., 2022). These new methods have been precipitated by the increased complexity in the output of models. Autoregressively generating a sequence which contains a label does not naturally provide a probability for the output being a valid pseudo-label. However, the outputs of LLMs tend to have more structure (for example, the generation of rationales, see Rajani et al. 2019; Nye et al. 2021; Wei et al. 2022) than traditional multinomial classification, allowing alternate confidence measures.

For example, Haluptzok et al. 2022 generate test cases in a programming language (unlabeled examples), then try to generate programs satisfying those test cases (predicted labels). Running the program can then identify which programs satisfy the test cases (allowing us to identify valid pseudo-labels). Zelikman et al. 2022 identify a similar approach, using reasoning tasks (for example, math problems) with numerical solutions as their unlabeled examples, generating potential solutions in natural language as their predicted labels, then keeping those solutions which reach the correct numerical solution as their pseudo-labels. Huang et al. 2022 provide a method closer to traditional self-training, by using self-consistency (Wang et al., 2022) to provide an explicit confidence score. Techniques have also been developed to provide the explicit probabilities with which LLMs believe their output to be correct (Kadavath et al., 2022) and which could be used with traditional confidence measures, although to our knowledge this has not yet been done.

However, large language models also provide methods for revising self-training beyond new choices of confidence measure. That is what we explore in this paper. Previous work on modifying self-training has looked at self-training using multiple classifiers. Typically, this co-training uses consensus in predictions as a confidence measure by training different models on different views of the same example (Blum and Mitchell, 1998; Miyato et al., 2017; Han et al., 2018). These "views" are different kinds of data about the same example (e.g. a webpage could be classified either by the text on that page, or by the text on all the pages linking to it). With LLMs, instead of using different sets of data about the same examples, we can use different methods available to the language model for solving the same problem.

## 3 METHOD

Figure 1 provides an illustrated overview of our method. We start with an unlabeled training dataset $D$ consisting of task examples $x_i \in D$, a Large Language Model (LLM), and $m$ distinct methods of solving a problem $M_1, M_2, ..., M_m$ using the LLM. For each example $x_i$ and each method $M_j$, we generate candidate solutions which can be considered predicted labels for the task. We then apply a confidence measure to identify the candidate solutions which are reliable pseudo-labels, allowing us to create a set of training examples. Finally, the training examples are used to train all $m$ methods by using the LLM to translate them from the original method used to produce the pseudo-label into the method being trained.

In this paper, we consider multi-method self-training with two methods: solving math problems via chain of thought prompting (Wei et al., 2022) (text), and solving math problems by writing a python function (Chen et al., 2021) (code). For our confidence measure, we check the final numerical answer produced by our model against the known numerical answer for the question. Although this guarantees correctness of the final numerical answer, it does not guarantee the correctness of the pseudo-label, as the model might generate the right answer by an incorrect chain of thought or incorrect code snippet. We make these specific choices of $m = 2$ methods, as well as the choice of specific confidence metric, as a simple test bed to demonstrate the effectiveness of multi-method self-training. In practice, multi-method self-training could be used with any self-training technique as long as the model being trained is capable of generating output through multiple methods and solutions from one method can be used to train another (see Appendix A for the prompts used in translation).

## 4  EXPERIMENTAL SETUP

### 4.1  TASKS & DATASETS.

We demonstrate the effectiveness of multi-method self-training on two types of tasks:

- **Arithmetic Reasoning.** We train our models to solve a diverse set of math word problems. The math problem datasets we use are SVAMP (Patel et al., 2021), GSM8K (Cobbe et al., 2021), MAWPS (Koncel-Kedziorski et al., 2016), and MathQA (Amini et al., 2019).
- **Out of Domain Tasks.** In addition to training and evaluating on math problem solving, we also evaluate (but do not train) on two other reasoning datasets: StrategyQA (Geva et al., 2021) and CommonSenseQA (Talmor et al., 2019).

### 4.2  MODEL.

In our experiments, we use the BLOOM large language model (Scao et al., 2022) with 176 billion parameters. The CoT and code prompts for each dataset are listed in Appendix A. We generate $k$ solutions for each problem in the training set, and only keep solutions whose numeric answers match the known numeric answer for the problem. We decode using nucleus sampling (Holtzman et al., 2019) with p=0.9 and a temperature of T=0.2. We train the model for 6 epochs with a learning rate of 1e-5 and a batch size of 32.

For each method, we compare our multi-method self-training approach to the following baselines:

- **BLOOM.** The baseline BLOOM model without any fine-tuning.
- **Single-Method Self-Training.** We train a model using single-method self-training. When evaluating the performance of MMST on text, we compare against single-method self-training on text alone. Similarly, when evaluating on code, we compare against single-method self-training on code alone. In both cases, we only use the confidence measure of matching the known numerical answer for the problem.

### 4.3  GENERATING SOLUTIONS.

All code snippets generated by BLOOM were in the python programming language. The response is generated in a function called `solution`. The code copied to the `globals` using the python `exec` function, and only the `solution` function is run. All other generated code is ignored. If multiple `solution` functions were generated, only the first one is run and all others are ignored. If the `solution` function produces an error or does not return a numerical value, the answer is considered wrong. We also use the autoflake package to remove any unnecessary code from the `solution` before training.

### 4.4  EVALUATION.

We evaluate our models on the held-out test sets for each of the arithmetic reasoning datasets (SVAMP, GSM8K, MAWPS, and MathQA). Additionally, we evaluate the models on the out-of-domain tasks (StrategyQA and CommonSenseQA) to measure potential improvements in reasoning capabilities. For evaluation, we report the percentage of correct answers generated by the model.

## 5  RESULTS

### 5.1  IMPROVING TEXT GENERATION

In our first experiment, we focus on generating solutions to math problems using Chain-of-Thought (CoT) prompting. We compare the performance of Bloom and single-method self-training (ST) to that of the MMST model. The MMST model is self-trained using the output from both chain of thought prompting and from code generation, whereas the base Bloom model has not been trained with any self-training method.

Table 1: Performance of BLOOM on mathematical reasoning datasets using chain-of-thought prompting. We compare the performance of BLOOM without finetuning, BLOOM finetuned using single-method self-training, and BLOOM trained using multi-method self-training.

| DATA SET | $BLOOM$ TEXT | $ST$ TEXT | $MMST$ TEXT |
|---|---|---|---|
| SVAMP | 28.6% | 40.1% | **52.2%** |
| GSM8K | 12.9% | 25.3% | **39.3%** |
| MAWPS | 70.0% | 95.4% | **100.0%** |
| MATHQA | 2.5% | 5.9% | **7.1%** |

Code generation is known to outperform language generation in math word problem solving (Pi et al., 2022; Gao et al., 2022). This allows us to verify that multi-method self-training can be used to improve the performance of a less-performant method using the output from a more performant method.

The results of this experiment can be seen in Table 1. Multi-method self-training leads to large improvements in math problem solving compared to both baselines using Chain-of-Thought prompting, and this improvement is seen on every dataset. This makes sense, as training data is added from code-generation, which achieves a higher performance on math problem solving than Chain-of-Thought prompting.

In addition to this automatic evaluation, we also provided the outputs of the BLOOM and MMST models to human annotators for subjective evaluation. All human annotators were professional computer scientists who volunteered to evaluate the the models. For each problem, the annotators were shown the annotation guidelines, along with two answers in a randomized order: one produced by Bloom and the other produced by the MMST model (see Appendix B). The annotators then selected the answer which they preferred, or "both" if the two outputs were considered equally good.

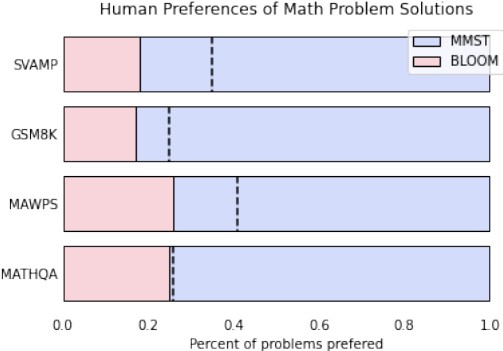

Figure 2: Human annotators were shown solutions to the same problem generated by BLOOM and MMST, and asked to select which they preferred. The above graph shows the proportion of problems for which the solution by BLOOM/MMST was preferred. The dashed line shows what percentage of the time each solution should be preferred if the determination was only made based on correctness.

The results from human evaluation can be found in Figure 2. The annotators typically preferred the output from the MMST model to that from the BLOOM model. It is notable that the margin by which annotators preferred the MMST model was greater than the margin by which it outperformed BLOOM in the automatic evaluation. This indicates the MMST improved not only the correctness of the model, but also the quality of the explanations generated by the model. This is notable because the model pseudolabels were only based on correctness when compared to the final numerical answer of each problem.

Table 2: Performance of BLOOM on mathematical reasoning datasets using code generation. We compare the performance of BLOOM without finetuning, BLOOM finetuned using single-method self-training, and BLOOM trained using multi-method self-training.

| Data set | $BLOOM$ Code | $ST$ Code | $MMST$ Code |
|---|---|---|---|
| SVAMP | 53.4% | 80.2% | **85.6%** |
| GSM8K | 32.5% | 46.9% | **52.6%** |
| MAWPS | **100.0%** | **100.0%** | **100.0%** |
| MATHQA | 7.2% | 14.5% | **14.8%** |

Table 3: The performance of multi-method self-training when the total amount of data is limited to the quantity of self-training data. We produce two models with two limits on data quantity. In the first ($MMST_{limited}$ TEXT), the quantity of data is limited to the data produced when self-training on text alone. In the second ($MMST_{limited}$ CODE). We report the performance of each model using the method from which its data limit came.

| Data set | $MMST_{limited}$ TEXT | $MMST_{limited}$ CODE |
|---|---|---|
| SVAMP | 45.3% | 81.1% |
| GSM8K | 30.4% | 48.0% |
| MAWPS | 100.0% | 100.0% |
| MATHQA | 6.8% | 12.2% |

## 5.2 IMPROVING CODE GENERATION

In our second experiment, we focus on generating solutions to math problems using code generation. We compare the performance of BLOOM and single-method self-training to that of the MMST model. Code generation outperforms Chain-of-Thought prompting in math word problem solving when using BLOOM without any finetuning. Therefore, if the MMST model outperforms the baselinse, it shows that multi-method self-training can improve the performance of a more performant method *using outputs from a less performant method.*

The results of this experiment are show in Table 2. Multi-method self-training leads to large performance improvements in math problem solving using code generation. This is a surprising result: how can outputs from a less performant method (such as text generation) be used to improve a more performant one (code generation)? Here, we outline several hypotheses.

### 5.2.1 DOES MULTI-METHOD SELF-TRAINING WORK BECAUSE OF DATA QUANTITY?

The first hypothesis is that multi-method self-training produces more training data. Instead of training on only examples from code generation, we can also train from examples of text generation. The larger number of training examples may lead to higher performance.

To test this hypothesis, we repeated the multi-method self-training process, but in each epoch, we randomly selected training examples without replacement until the number of training examples used in that epoch was equal to the number of examples generated when self-training with a single method.

The results of this ablations analysis are reported in table 3. Multi-method self-training, even on limited quantities of data, results in a marked increase in performance for solving arithmetic problems using text. Accordingly, a majority of the improvement results from the use of multiple methods, as opposed to the larger quantity of data. A similar pattern is observed when training with limited amounts of data using the code method. However, the degree of increase is smaller, and on MATHQA, the model trained on a limited amount of data under-performs single-method self-training on code alone.

### 5.2.2 DOES MULTI-METHOD SELF-TRAINING WORK BECAUSE OF ANTI-CORRELATION BETWEEN METHODS?

The second hypothesis is that the gains are derived from a distributional shift in the kinds of problems which the methods can solve. Many of the problems solved by text generation are not solved by code generation. Thus, although the overall performance of code generation may be higher than text generation, there are capabilities which text generation has that code generation does not. It may be that these are the capabilities learned during multi-method self-training.

Another way of looking at this improvement is through the lens of covariance. Let's say that we have $m$ methods ($M_1, ..., M_m$) for solving a problem. For any given problem, each method $M_i$ has an expected performance $\mu_i$, and the actual performance may differ by some noise factor $\epsilon_i$, so the actual performance is $X_i = \mu_i + \epsilon_i$. Finally, the model is learning from the various examples, making the model's performance a function of the various $X_i$. Let's call this function $\mathcal{A}$ (the *aggregation* function).

In multi-method self-training, the properties of $X_i$ and $\mathcal{A}$ are potentially complex and unknown. However, we can get an intuition by looking at a simple case. Let's say that there is no difference in mean performance between the different methods. Can multi-method self-training still result in an improvement? If $X_i$ are i.i.d. normal, and $\mathcal{A}$ is the maximum function, then yes (i.e. $E[X_i] < E[max_i(X_i)]$). In that case, the performance of the model trained with multi-method self-training is given by the extreme value distribution, which has higher mean than $X_i$.

Is it reasonable to say that the aggregation function in multi-method self-training is comparable to the maximum function? The use of a confidence metric suggests yes. Assuming the confidence metric is strongly correlated with quality (as it is in the case of correctness in solving a math word problem), then we are training a model only on examples above a certain threshold of quality. If we generate $m$ answers, and only one is above the threshold, then $\mathcal{A}$ is similar to the maximum function.

This analogy allows us to provide an intuition for the impact of covariance on the improvements from multi-method self-training. If the correlation between two methods is high, then the improvement from multi-method self-training that results from the aggregation mechanism would be low. For example, if the correlation is 1, the maximum of several equal values $X_i = X$ is still that value $X$. On the other hand, if the performance of the different methods is anti-correlated (i.e. the methods perform well on different tasks, or the covariance is -1), the expected improvement from multi-method self-training is large, because the difference between the best and worst methods grows larger.

We can also extend our analysis to aggregation functions beyond the maximum function. Consider the broader class of convex functions. A function $f$ is convex if, for any two points $x$ and $y$, and any $t \in [0, 1]$, we have $f(tx + (1 - t)y) \leq tf(x) + (1 - t)f(y)$.

Jensen's inequality states that for any convex function $f$ and any random variable $X$, the expectation of the function over the random variable is greater than or equal to the function of the expectation of the random variable, i.e., $E[f(X)] \geq f(E[X])$. This inequality provides insights into the behavior of convex functions when applied to random variables, which can be helpful in understanding the performance of multi-method self-training models.

In our case, the maximum function serves as a specific example of a convex function. When applied to the random variables $X_i$, which represent the performance of different methods, Jensen's inequality implies that the expected performance of the model trained with multi-method self-training (i.e., $E[\mathcal{A}(X_i)]$) is greater than or equal to the maximum of the expected performance of individual methods (i.e., $\mathcal{A}(E[X_i])$), and our analysis with the maximum function extends to any convex function.

For instance, one could consider functions that aggregate the performance of different methods in a non-linear manner, such as $\mathcal{A}(X_1, \ldots, X_m) = (w_1 X_1^2 + \cdots + w_m X_m^2)^{1/2}$, where $w_i$ are non-negative weights that sum to one. In this case, $\mathcal{A}$ is a convex function, as it is a weighted quadratic mean of the $X_i$s. Applying Jensen's inequality, we find that the expected performance of a multi-method self-training model using such an aggregation function would be greater than or equal to the quadratic mean of the expected performance of individual methods, i.e., $E[\mathcal{A}(X_i)] \geq \mathcal{A}(E[X_i])$. The fact that any convex aggregation function improves performance (and possibly many

Table 4: The correlation between text and code answers generated for each dataset. Each method was modeled as a binary variable and each example in the dataset was considered an observation: 0 if the method came up with an incorrect final answer to the problem or 1 if the method came up with a correct final answer to the problem. We report both the correlation considering all examples in each dataset, and also the correlation between examples with at least one positive pseudolabel.

| CORRELATION | ALL | POSITIVE PSEUDOLABELS |
|---|---|---|
| SVAMP | 0.205 | -0.437 |
| GSM8K | 0.226 | -0.552 |
| MAWPS | 0.174 | -0.333 |
| MATHQA | 0.193 | -0.731 |

Table 5: Performance of BLOOM on out of domain reasoning datasets before and after multi-method self-training. Although text generation out-performs code generation on these datasets in the base BLOOM model, text generation performance still improves with multi-method self-training.

| DATA SET | $BLOOM$ CODE | $BLOOM$ TEXT | $MMST$ TEXT |
|---|---|---|---|
| STRATEGYQA | 45.4% | 61.3% | 68.2% |
| COMMMONSENSEQA | 30.7% | 49.3% | 59.6% |

non-convex functions besides) increases the plausibility that improvements from multi-method self-training benefit from methods with low correlation.

The correlation between the two methods when using BLOOM on each dataset (reported in table 4) provide preliminary evidence in this direction. Although we cannot access the quality of each solution directly, we can use the pseudo-labels as a proxy for quality. Furthermore, the model is not trained on all examples from each dataset, only on the positive pseudolabels, so we also report the correlation among examples where at least one model produces a positive pseudolabel.

Datasets which had a more negative correlation among the pseudolabels saw a greater improvement from MMST, with the exception of MathQA. This suggests that anti-correlation between methods is likely to produce greater performance when using MMST, but that other factors also influence the final performance. The fact that MathQA had the lowest initial performance and the smallest quantity of training data also suggest additional factors that may be relevant in determining the effectiveness of MMST.

These hypotheses are by no means exhaustive, but provide potential explanations for the counter-intuitive result that the output from a less performant method can be used to improve that of a more performant method. More research would be required to determine the precise cause of these improvements.

## 5.3 IMPROVING OUT OF DOMAIN TASKS

In our third experiment, we focus on generating solutions to strategy and commonsense questions. We compare the performance of Bloom using Chain-of-Thought (CoT) prompting and code generation, as well as to the MMST model using CoT.

The results are in table 5. In both tasks, CoT outperforms code generation with BLOOM, but the MMST model outperforms both. This indicates that multi-method self-training can improve out-of-domain tasks, as the MMST model was trained only on math problem solving. We hypothesize that this is caused by improvements of abilities which are upstream of both math problem solving and commonsense question answering (for example, the generation of rationales).

## 6 CONCLUSION & FUTURE WORK

In this paper, we demonstrated that multi-method self-training is capable of improving Large Language Model (LLM) performance for the less performant method, the more performant method, and for out of domain tasks. Experiments using a 176B parameter LLM showed that our approach

improves accuracy scores on math word problem solving by as much as 30%. Furthermore, we analyzed why multi-method self-training works via ablation.

We see two clear avenues for future work. The first avenue is in the application of multi-method self-training to multi-modal models. Prior work has shown that creating multi-modal models allows for applications to a much larger set of problems, from document intelligence (Xu et al., 2021), to image generation (Ramesh et al., 2022), to robotics (Driess et al., 2023). These multi-modal models are also able to solve problems through a more diverse set of methods (Huang et al., 2023). As multi-modal models become more important going forward, multi-method self-training with such models may be a useful technique. The second avenue is in better understanding multi-method self-training (e.g. what kinds of tasks does multi-method self-training work well for, can we automatically identify what the multiple methods should be, etc.).

However, we think there is also another interesting avenue signaled by our results. LLMs have traditionally been trained using self-supervised learning. This trains them to be next-token predictors, or simulators (as has been suggested by practitioners such as Janus 2022). Recent work has shown that you can get enormous gains in performance on many tasks by changing the way in which we train these models. For example, instruction-finetuning (Sanh et al., 2021; Wei et al., 2021; Chung et al., 2022), Reinforcement Learning from Human Feedback (RLHF) (Stiennon et al., 2020; Ouyang et al., 2022), Reinforcement Learning from AI Feedback (RLAIF) (Bai et al., 2022), and various self-training methods (Haluptzok et al., 2022; Zelikman et al., 2022; Huang et al., 2022). This makes sense – recent work training models like Chinchilla (Hoffmann et al., 2022) and Minerva (Lewkowycz et al., 2022) suggest that the primary bottlenecks in model performance are the quantity and quality of data available to the model. Many of these methods serve as means of providing large quantities of high-quality data to a model. However, we have only begun to study these techniques; there are likely other simple and effective training methods waiting to be discovered, as we hope our work shows. We encourage the exploration of this space – the space of new training methods for LLMs – as a fruitful domain for future work.

## 7  LIMITATIONS

We would like to note two different sets of limiations in our paper.

The first set of limitations we would like to mention are limitations of MMST. An assumption of MMST is that each method produces artifacts which can be used to train the other. This is a weak assumption, as the final labels produced for a task should not vary by method. For example, we could try to classify webpages into spam or non-spam by images or text – the labels would be spam or non-spam either way. However, the method is likely to be more effective if more information can be transferred between the methods. LLMs can easily convert text to code and vice versa, but this may be more difficult for other methods in other tasks. Similarly the specifics of the methods used can have a large impact on the model (e.g. the prompt can influence performance significantly, see Appendix C). In this paper, the prompts were hand-crafted, which is not scalable. However, this limitation may be alleviated by using prompt-optimization methods during training. Finally, self-training and related methods such as reinforcement learning are known to suffer from training instability (Henderson et al., 2017; Sohn et al., 2020).

The second set of limitations we would like to mention are limitations of our analysis. We analyze MMST using one model (BLOOM 176B) and two types of tasks (math problem solving and other reasoning tasks). While we test on multiple tasks of each type, both of these factors may impact the results achieved with MMST and we do not vary them significantly. As such, our paper should be considered more an existence proof for the results that can be achieved using MMST, as opposed to a claim that the method will work with all models and all tasks.

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

# A  PROMPTS

## A.1  PROMPTS USED FOR MATH PROBLEM SOLVING

**Solving Math Problems With Text**

```
There are 15 trees in the grove. Grove workers will plant trees in the
    grove today. After they are done, there will be 21 trees. How many
    trees did the grove workers plant today?
Let's think step-by-step.
There are 15 trees originally. Then there were 21 trees after some more
    were planted. So there must have been 21 - 15 = 6. The answer is 6.

If there are 3 cars in the parking lot and 2 more cars arrive, how many
    cars are in the parking lot?
Let's think step-by-step.
There are originally 3 cars. 2 more cars arrive. 3 + 2 = 5. The answer is
    5.

Leah had 32 chocolates and her sister had 42. If they ate 35, how many
    pieces do they have left in total?
Let's think step-by-step.
Originally, Leah had 32 chocolates. Her sister had 42. So in total they
    had 32 + 42 = 74. After eating 35, they had 74 - 35 = 39. The answer
    is 39.

Jason had 20 lollipops. He gave Denny some lollipops. Now Jason has 12
    lollipops. How many lollipops did Jason give to Denny?
Let's think step-by-step.
Jason started with 20 lollipops. Then he had 12 after giving some to
    Denny. So he gave Denny 20 - 12 = 8. The answer is 8.

Shawn has five toys. For Christmas, he got two toys each from his mom and
     dad. How many toys does he have now?
Let's think step-by-step.
Shawn started with 5 toys. If he got 2 toys each from his mom and dad,
    then that is 4 more toys. 5 + 4 = 9. The answer is 9.

There were nine computers in the server room. Five more computers were
    installed each day, from monday to thursday. How many computers are
    now in the server room?
Let's think step-by-step.
There were originally 9 computers. For each of 4 days, 5 more computers
    were added. So 5 * 4 = 20 computers were added. 9 + 20 is 29. The
    answer is 29.

Michael had 58 golf balls. On tuesday, he lost 23 golf balls. On
    wednesday, he lost 2 more. How many golf balls did he have at the end
     of wednesday?
Let's think step-by-step.
Michael started with 58 golf balls. After losing 23 on tuesday, he had 58
     - 23 = 35. After losing 2 more, he had 35 - 2 = 33 golf balls. The
    answer is 33.

Olivia has $23. She bought five bagels for $3 each. How much money does
    she have left?
Let's think step-by-step.
Olivia had 23 dollars. 5 bagels for 3 dollars each will be 5 x 3 = 15
    dollars. So she has 23 - 15 dollars left. 23 - 15 is 8. The answer is
     8.

{question}
Let's think step-by-step.
{model output}
```

**Solving Math Problems With Code**

```
"""
Write a function which computes and returns the solution to the following
    word problem:
There are 15 trees in the grove. Grove workers will plant trees in the
    grove today. After they are done, there will be 21 trees. How many
    trees did the grove workers plant today?
"""
def solution():
  # Given
  start_trees = 15
  end_trees = 21

  # Return
  return end_trees - start_trees

"""
Write a function which computes and returns the solution to the following
    word problem:
If there are 3 cars in the parking lot and 2 more cars arrive, how many
    cars are in the parking lot?
"""
def solution():
  # Given
  cars = 3
  cars += 2

  # How many cars are in the parking lot?
  return cars

"""
Write a function which computes and returns the solution to the following
    word problem:
Leah had 32 chocolates and her sister had 42. If they ate 35, how many
    pieces do they have left in total?
"""
def solution():
  # Given
  chocolates = {{"leah": 32, "sister": 42}}
  chocolates["total"] = sum(chocolates.values())
  chocolates["eaten"] = 35

  # How many pieces do they have left in total?
  return chocolates["total"] - chocolates["eaten"]

"""
Write a function which computes and returns the solution to the following
    word problem:
Jason had 20 lollipops. He gave Denny some lollipops. Now Jason has 12
    lollipops. How many lollipops did Jason give to Denny?
"""
def solution():
  # Given
  jason_start = 20
  jason_end = 12
  denny = jason_start - jason_end

  # How many lollipops did Jason give to Denny?
  return denny

"""
Write a function which computes and returns the solution to the following
    word problem:
```

```
Shawn has five toys. For Christmas, he got two toys each from his mom and
    dad. How many toys does he have now?
"""
def solution():
  # Given
  shawn_start = 5
  mom_gave = 2
  dad_gave = 2

  # How many toys does he have now?
  return shawn_start + mom_gave + dad_gave

"""
Write a function which computes and returns the solution to the following
    word problem:
There were nine computers in the server room. Five more computers were
    installed each day, from monday to thursday. How many computers are
    now in the server room?
"""
def solution():
  # Given
  computers_start = 9
  computers_per_day = 5
  # Thursday - Monday = 4 days
  days = 4

  # How many computers are now in the server room?
  return computers_start + computers_per_day * days

"""
Write a function which computes and returns the solution to the following
    word problem:
Michael had 58 golf balls. On tuesday, he lost 23 golf balls. On
    wednesday, he lost 2 more. How many golf balls did he have at the end
    of wednesday?
"""
def solution():
  # Given
  balls = 58
  balls -= 23
  balls -= 2

  # How many golf balls did he have at the end of wednesday?
  return balls

"""
Write a function which computes and returns the solution to the following
    word problem:
Olivia has $23. She bought five bagels for $3 each. How much money does
    she have left?
"""
def solution():
  # Given
  olivia_money = 23
  num_bagels = 5
  cost_of_bagel = 3

  # How much money does she have left?
  return olivia_money - num_bagels * cost_of_bagel

"""
Write a function which computes and returns the solution to the following
    word problem:
{question}
"""
```

```
def solution:
    # Given
    {model output}
```

## A.2 PROMPTS USED FOR TRANSLATING BETWEEN METHODS

**Translating Between Methods**

Q: There are 15 trees in the grove. Grove workers will plant trees in the
    grove today. After they are done, there will be 21 trees. How many
    trees did the grove workers plant today?
A: There are 15 trees originally. Then there were 21 trees after some
    more were planted. So there must have been 21 − 15 = 6. The answer is
    6.
Code:
```
def solution():
    # Given
    start_trees = 15
    end_trees = 21
    # Return
    return end_trees − start_trees
```

Q: If there are 3 cars in the parking lot and 2 more cars arrive, how
    many cars are in the parking lot?
A: There are originally 3 cars. 2 more cars arrive. 3 + 2 = 5. The answer
    is 5.
Code:
```
def solution():
    # Given
    cars = 3
    cars += 2

    # How many cars are in the parking lot?
    return cars
```

Q: Leah had 32 chocolates and her sister had 42. If they ate 35, how many
    pieces do they have left in total?
A: Originally, Leah had 32 chocolates. Her sister had 42. So in total
    they had 32 + 42 = 74. After eating 35, they had 74 − 35 = 39. The
    answer is 39.
Code:
```
def solution():
    # Given
    chocolates = {{"leah": 32, "sister": 42}}
    chocolates["total"] = sum(chocolates.values())
    chocolates["eaten"] = 35

    # How many pieces do they have left in total?
    return chocolates["total"] − chocolates["eaten"]
```

Q: Jason had 20 lollipops. He gave Denny some lollipops. Now Jason has 12
    lollipops. How many lollipops did Jason give to Denny?
A: Jason started with 20 lollipops. Then he had 12 after giving some to
    Denny. So he gave Denny 20 − 12 = 8. The answer is 8.
Code:
```
def solution():
    # Given
    jason_start = 20
    jason_end = 12
    denny = jason_start − jason_end

    # How many lollipops did Jason give to Denny?
    return denny
```

Q: Shawn has five toys. For Christmas, he got two toys each from his mom
    and dad. How many toys does he have now?
A: Shawn started with 5 toys. If he got 2 toys each from his mom and dad,
    then that is 4 more toys. 5 + 4 = 9. The answer is 9.
Code:
```
def solution():
    # Given
    shawn_start = 5
    mom_gave = 2
    dad_gave = 2

    # How many toys does he have now?
    return shawn_start + mom_gave + dad_gave
```

Q: There were nine computers in the server room. Five more computers were
    installed each day, from monday to thursday. How many computers are
    now in the server room?
A: There were originally 9 computers. For each of 4 days, 5 more
    computers were added. So 5 * 4 = 20 computers were added. 9 + 20 is
    29. The answer is 29.
Code:
```
def solution():
    # Given
    computers_start = 9
    computers_per_day = 5
    # Thursday - Monday = 4 days
    days = 4

    # How many computers are now in the
    server room?
    return computers_start + computers_per_day * days
```

Q: Michael had 58 golf balls. On tuesday, he lost 23 golf balls. On
    wednesday, he lost 2 more. How many golf balls did he have at the end
    of wednesday?
A: Michael started with 58 golf balls. After losing 23 on tuesday, he had
    58 - 23 = 35. After losing 2 more, he had 35 - 2 = 33 golf balls.
    The answer is 33.
Code:
```
def solution():
    # Given
    balls = 58
    balls -= 23
    balls -= 2

    # How many golf balls did he have at the end of wednesday?
    return balls
```

Q: Olivia has $23. She bought five bagels for $3 each. How much money
    does she have left?
A: Olivia had 23 dollars. 5 bagels for 3 dollars each will be 5 x 3 = 15
    dollars. So she has 23 - 15 dollars left. 23 - 15 is 8. The answer is
    8.
Code:
```
def solution():
    # Given
    olivia_money = 23
    num_bagels = 5
    cost_of_bagel = 3

    # How much money does she have left?
    return olivia_money - num_bagels * cost_of_bagel
```

Q: {question}

```
A: {answer}
Code:
{model_output}
```

Note: when translating from code to CoT, the order of "A:" and "Code:" above were switched.

## A.3 PROMPTS USED FOR ABLATION ANALYSIS

The prompts used for ablation analysis were k-shot, similar to the other prompts. However, for succinctness of expression, we show a zero-shot version, as all the examples used are identical to those in the previous prompts.

---

**CoT (Code)**

```
```python
"""
{question}
"""
{model output}
```
So the answer is {model output}
```

---

**+ Computation**

```
"""
Write a function which computes and
returns the solution to the following
word problem:
{question}
The function must return a single
numerical value. It cannot print the
answer.
"""
def solution(): {model output}
```

---

**+ Extract Quantities**

```
"""
Write a function which computes and
returns the solution to the following
word problem:
{question}
The function must return a single
numerical value. It cannot print the
answer.
"""
def solution():
  # Given
  {model output}
```

## B  ANNOTATION

Below are two solutions to the following math problem

Each pack of dvds costs 76 dollars. If there is a discount of 25 dollars on each pack, how much do you have to pay to buy each pack?

Please choose the better solution.
A correct answer should always be preferred to a incorrect one.
A correct explanation should always be preferred to an incorrect one.
A solution with a clearer explanation which would be more helpful to a learner should be preferred over a less clear answer.

**A**  Given the cost of a pack is 76 dollars, we can subtract the discount to get the final price.

76 - 25 = 51.

So the answer is 51.

**B**  1. The price is initially $76 (start_price = 76).
2. The discount is $25 (discount = 25).
3. We can get the final price by subtracting the discount from the start price (start_price - discount = 76 - 25 = 51).

So the answer is 51.

| Prefer **A** | **A** and **B** are equal | Prefer **B** |

Figure 3: The annotation interface used by annotators in section 5

# C  WHY DOES CODE GENERATION OUTPERFORM TEXT GENERATION FOR MATH PROBLEM SOLVING?

Table 6: The performance of BLOOM on SVAMP as we ablate the prompt.

|  | CoT (TEXT) | CoT (CODE) | + CALC | + QUANTITIES | + RELATIONSHIPS |
|---|---|---|---|---|---|
| ACCURACY | 28.6% | 28.3% | 36.6% | 43.5% | 53.4% |
| IMPROVEMENT | – | – | +8.3% | +6.9% | +9.9% |

Up to this point, we have established that multi-method self-training can significantly outperform single-method self-training and that the effectiveness of multi-method self-training depends on factors such as the quantity of data generated and the degree of correlation between the solutions provided by different methods (although these factors may not be exhaustive). Another area of interest when applying multi-method self-training is understanding why different methods provide different outputs. This can help to identify the most effective methods to use in multi-method self-training.

Although we do not have an exhaustive means of determining the differences between methods, we propose to understand the differences between text generation and code generation by ablation analysis; namely, removing parts of the code prompt until we achieve characteristics similar to the text prompt. We hope that this might provide some guidance on what methods can be used effectively with multi-method self-training.

We ablate the code prompt used for multi-method self-training along the steps in (Jie et al., 2022). Jie et al. formulate math word problem solving as a complex relation extraction task with three steps: 1) extract quantities from the problem 2) extract relationships between those quantities (where relationships are operations involving multiple quantities), and 3) use the extracted quantities and relationships to do a computation. Our first prompt removes the explicit extraction of relationships, our second prompt additionally removes the explicit extraction of quantities, and finally we remove the explicit computation by asking the LLM to provide an answer from the code that it generated to solve the problem. Because this required manual analysis, we only report results on SVAMP.

The results of this ablation analysis are reported in table 6. There are two things worth noting about these results. The first is that naive application of code does not out-perform chain of thought prompting. Second, the improved performance of the final prompt is the result of many cumulative improvements, rather than a single large improvement. That suggests that, while multi-method self-training can be used with any two methods, a larger benefit can be accrued by understanding the differenes between the methods. We suspect that this will be true of multi-method self-training applied to other problems as well, and that a more general understanding of the differences between methods will allow for more effective application of multi-method self-training.

