# OpenReview forum: "Multi-Method Self-Training: Improving Code Generation With Text, And Vice Versa"
_ICLR.cc/2024/Conference — Submitted to ICLR 2024_

### Official Review · Reviewer_jsGy · 2023-10-31

**Soundness:** 3 good
**Presentation:** 4 excellent
**Contribution:** 3 good
**Rating:** 6
**Confidence:** 3

**Summary:**

This work introduces a technique to self-train LLMs by leveraging their capabilities to (1) solve problems using multiple methods (code and text) and (2) translate solutions between methods. This multi-method self-training (MMST) solution is shown to improve human-rated performance for the BLOOM language model on both arithmetic tasks on which the method can be used to self-train, and on other tasks which benefit from the self-finetuned model's improved reasoning capabilities. In addition, the paper formulates 2 hypotheses on why this method works and finds that (1) MMST's improved performance is not solely linked to additional training examples, and (2) MMST benefits from leveraging methods that are anti-correlated.

**Strengths:**

* Clarity: the paper provides relevant context and motivation. Experiments are clearly described and the authors' reasoning is easy to follow.

* Quality: the work is well executed and its conclusions are clear. One drawback of the current presentation of results is that they do not give a clear sense of how they compare to other self-improvement techniques that have published promising results on arithmetic tasks.

* Originality: while a number of recent works have proposed techniques for models to self-improve, the proposed multi-domain technique and its leverage of translation capabilities across methods are a novel and relevant way to leverage the diversity of tasks LLMs can handle.

* Significance: while the results look promising, their significance is hard to grasp without a clear head-to-head comparison with preexisting methods.

**Weaknesses:**

* As the authors clearly state, multiple papers have been published previously on self-improved language models. The paper could include an experimental comparison with them and a discussion of the relative benefits of each approach. In particular, the following works could be discussed in more detail: (Li et al, 2022, "Making Large Language Models Better Reasoners with Step-Aware Verifier"; Huang et al., 2022, "Large Language Models Can Self-Improve")

**Questions:**

* The training process could be described more precisely: how many training examples is the model fine-tuned on? Are the training examples re-generated at each epoch?

* Do out-of-domain tasks actually benefit from the multi-method aspect? It may be worth comparing the results to single-method fine-tuning.

* How would the overall results compare to other self-improved language models, e.g. based on chain of thought? A comparison with Li et al, 2022, "Making Large Language Models Better Reasoners with Step-Aware Verifier", and Huang et al., 2022, "Large Language Models Can Self-Improve", would give more credence to the proposed approach.

---

### Official Review · Reviewer_kd49 · 2023-11-01

**Soundness:** 3 good
**Presentation:** 2 fair
**Contribution:** 2 fair
**Rating:** 3
**Confidence:** 4

**Summary:**

This paper introduces Multi-Method Self-Training (MMST) to enhance the performance of LLMs where one method (text generation method) is trained on the filtered output of the other (code generation model) and vice versa. Empirical results show improved performance on solving math problems (MathQA, SVAMP, GSM8K, MAWPS arithmetic reasoning datasets) and out of domain tasks (StrategyQA, CommonSenseQA datasets) with the BLOOM model. Interesting ablation studies help understand why the MMST works.

**Strengths:**

- MMST effectively enhances the capabilities of both less performant and more performant methods, making the model more versatile and capable.
- The proposed approach leverages multiple methods for solving a given problem, thus enhancing the strengths and mitigating the weaknesses of each method.
- Ablation studies based on increased data quantity and anti-correlation between methods are interesting.

**Weaknesses:**

- Paper could be improved in terms of writing structure and quality. A more structured and intuitive flow could enhance the paper's readability. The presentation of the paper can also be somewhat dense and succinct.
- The paper mainly focuses on math problem solving and out-of-domain reasoning tasks. It would be valuable to explore MMST to a broader range of tasks to assess its generalizability. It seems that this strategy would be applicable to limited tasks like arithmetic reasoning which could also be solved by formulating the task as generating code. Could the authors justify this?
- It seems the relevant baselines haven’t been used like GPT-3.5/4 or open-sourced LLMs like LLaMA-2 where the field has evolved a lot in recent years.
- Tables could be better formatted with \hline in latex for better readability. Also please use a high resolution image for Figure 2.

**Questions:**

- Could the authors confirm if a single BLOOM model was used for code generation and text generation or two different BLOOM models were used? Is there any curriculum used for training on the two different tasks?
- It seems that code generation BLOOM models consistently outperform the text generation method for arithmetic problems. Do authors have an intuition why this would be the case? In this regard, the performance of recent LLMs (GPT/LLaMA/CodeLLaMA) could help justify and strengthen the observation.
- Could it be confirmed if the corresponding code be made publicly available for reproducibility and reliability?

Suggestions/Comments:
Table 3 caption: . We -> second, we report

---

### Official Review · Reviewer_uxjq · 2023-11-01

**Soundness:** 3 good
**Presentation:** 1 poor
**Contribution:** 2 fair
**Rating:** 5
**Confidence:** 3

**Summary:**

The paper proposes multi-method self-training (MMST) for large language models. Given the multi-method capability of current large language models, the approach translates the self-generated answers from one method to another, and then employs the translated text for training in a co-training style. The paper focuses on arithmetic reasoning tasks with two methods: solving via chain of thought prompting and solving via writing code. Through extensive experiments, the authors demonstrate that multi-method self-training improves the trained model for math solving in both chain-of-thought and coding. Besides, the authors conduct ablation studies for two hypotheses: MMST works because of data quantity, and MMST works because of anti-correlation between methods. Moreover, they also observe improvements on out-of-domain tasks, generating solutions to strategy and commonsense questions.

**Strengths:**

- The proposed method is simple and effective for improving arithmetic reasoning performance. The results demonstrate that with multi-method self-training, the MMST-trained BLOOM model outperforms BLOOM and self-trained BLOOM  on four arithmetic reasoning datasets. The improvement is consistent for both chain of thought inference and writing codes.
- The paper has a counterintuitive finding that leveraging the self-labeled data from a less performant method can still improve the performance of the more performant method.

**Weaknesses:**

- The technical contribution is limited: generating pseudo training data with two different prompts as two different methods. It is necessary to conduct experiments on diverse code generation tasks, otherwise, the paper should narrow its scope to arithmetic reasoning.
- The writing needs to be improved. The paper is hastily written and a little difficult to follow. The method section is quite brief but section 5.2 is verbose.
- The paper lacks an in-depth analysis of improvement on out-of-domain tasks.

**Questions:**

What is the main takeaway of Section 5.2.2?

---

### Official Review · Reviewer_4TSd · 2023-11-01

**Soundness:** 3 good
**Presentation:** 3 good
**Contribution:** 3 good
**Rating:** 5
**Confidence:** 3

**Summary:**

The paper proposes a multi-method self-training to improve LLMs performance for arithmetic reasoning by designing the problem as math via chain of thought prompting and writing a python function. They demonstrate the effectiveness of the model using the BLOOM 176B model. They demonstrate that multi method self-training can improve out-of-domain tasks which are related to the self-training task.

**Strengths:**

- The proposed method is very useful in practice when we only have unlabeled data, and the method improves the model's performance before applying any finetuning.
- The evaluation is done on a diverse set of tasks on both in-domain and out-of-domain tasks.

**Weaknesses:**

- Minimal novelty. The idea is to combine two different self-training methods into a single method.
- Lack of baseline comparison. Please include any existing model performance in Tables 1 and 2 from the recent SOTA for completeness.
- The motivation is unclear as to why the authors chose only two multi-method approaches.
- The paper requires more technical and evaluation details (e.g., the metric shown in the results Table).

**Questions:**

Questions:
- How's the trend on smaller models? Do you find them the same trend?
- Do you have any insights on how the model can achieve 100% on MAWPS?
- It is unclear to me which among two methods is more useful.

Suggestions:
- Please use consistent capitalization (e.g., Bloom -> BLOOM)
- Add more details about the evaluation metric on the Table caption
- For presentation, it will be easier to read with borders.

---

### Author Response · Authors · 2023-11-23

We appreciate the reviewers' critiques and suggestions for this paper. Unfortunately, due to external circumstance, we have been unable to take the time to thoroughly examine and address these reviews during the discussion period. As such, we will take into account the recommendations here and re-submit to another venue in the future.

---

### Meta-Review · Area_Chair_1EkS · 2023-12-06

**Metareview:**

The paper presents a novel approach to self-training large language models (LLMs) by leveraging their ability to solve problems using multiple methods (prompts or instructions) and translate solutions between methods. The reviewers appreciated the clarity of the paper and the extensive experiments conducted to validate the proposed approach. The paper also provides interesting ablation studies.

However, the reviewers raised several concerns. The novelty of the paper was questioned, with some reviewers suggesting that the paper essentially combines two different self-training methods into a single method. The paper was also criticized for its lack of baseline comparison, with reviewers suggesting that the authors should include the performance of existing models for completeness.

The paper's focus on arithmetic reasoning tasks was also a point of contention, with some reviewers suggesting that the authors should conduct experiments on a diverse range of code generation tasks.

**Justification For Why Not Higher Score:**

See meta review

**Justification For Why Not Lower Score:**

See meta review

---

### Decision · Program_Chairs · 2024-01-16

Reject